# Predicting host taxonomic information from viral genomes: A comparison of feature representations

**Francesca Young**[1], **Simon Rogers**[2], **David L. Robertson**[1]*

**1** MRC-University of Glasgow Centre For Virus Research, Glasgow, United Kingdom, **2** School of Computing Science, University of Glasgow, Glasgow, United Kingdom

* david.l.robertson@glasgow.ac.uk

**Data Availability Statement:** All relevant data are within the manuscript and its Supporting Information files.

**Funding:** FY is supported by a studentship from the Medical Research Council (MRC). DLR is funded by the MRC (MC_UU_1201412). The

## Abstract

The rise in metagenomics has led to an exponential growth in virus discovery. However, the majority of these new virus sequences have no assigned host. Current machine learning approaches to predicting virus host interactions have a tendency to focus on nucleotide features, ignoring other representations of genomic information. Here we investigate the predictive potential of features generated from four different 'levels' of viral genome representation: nucleotide, amino acid, amino acid properties and protein domains. This more fully exploits the biological information present in the virus genomes. Over a hundred and eighty binary datasets for infecting versus non-infecting viruses at all taxonomic ranks of both eukaryote and prokaryote hosts were compiled. The viral genomes were converted into the four different levels of genome representation and twenty feature sets were generated by extracting k-mer compositions and predicted protein domains. We trained and tested Support Vector Machine, SVM, classifiers to compare the predictive capacity of each of these feature sets for each dataset. Our results show that all levels of genome representation are consistently predictive of host taxonomy and that prediction k-mer composition improves with increasing k-mer length for all k-mer based features. Using a phylogenetically aware holdout method, we demonstrate that the predictive feature sets contain signals reflecting both the evolutionary relationship between the viruses infecting related hosts, and host-mimicry. Our results demonstrate that incorporating a range of complementary features, generated purely from virus genome sequences, leads to improved accuracy for a range of virus host prediction tasks enabling computational assignment of host taxonomic information.

## Author summary

Elucidating the host of a newly identified virus species is an important challenge, with applications from knowing the source species of a newly emerged pathogen to understanding the bacteriophage-host relationships within the microbiome of any of earth's ecosystems. Current high throughput methods used to identify viruses within biological

funders had no role in study design, data collection and analysis, decision to publish, or preparation of the manuscript.

**Competing interests:** The authors have declared that no competing interests exist.

or environmental samples have resulted in an unprecedented increase in virus discovery. However, for the majority of these virus genomes the host species/taxonomic classification remains unknown. To address this gap in our knowledge there is a need for fast, accurate computational methods for the assignment of putative host taxonomic information. Machine learning is an ideal approach but to maximise predictive accuracy the viral genomes need to be represented in a format (sets of features) that makes the discriminative information available to the machine learning algorithm. Here, we compare different types of features derived from the same viral genomes for their ability to predict host information. Our results demonstrate that all these feature sets are predictive of host taxonomy and when combined have the potential to improve accuracy over the use of individual feature sets across many virus host prediction applications.

## Introduction

Determining which virus infects which host species is currently a major challenge in virology. Knowledge of virus-host infectivity is essential to understanding the impact that viruses have on cellular life and the key roles they play as an integral part of all earth's ecosystems, from our own microbiome [1] to the marine environment where they contribute to regulation of the biogeochemical cycles [2,3], and as animal and plant pathogens. Advances in metagenomics have led to a rapid expansion in virus discovery with more than half of all known viral genomes being deposited in databases in the last two years [4]. This growth in data is the first step towards cataloguing of the earth's virosphere. However, the indiscriminate nature of metagenomics results in the majority of these new viruses having no identified host. For example, there are now over 700 000 viral genomes in the IMG/VR databases of which less than 5% have an associated host [5,6]. Currently, there are no high-throughput methods available to make reliable virus-host associations and as such we are unable to keep up with the rapid pace of viral discovery. Fast, accurate computational tools are thus urgently needed to annotate these new viral genomes with host taxon information.

Computational approaches to virus host prediction fall into four broad strategies: searching for homologous sub-sequences in the hosts, such as prophage [5] or CRISPR-Cas spacers [7]; looking for co-abundance between virus and host [8]; distance based metrics of oligo-nucleotide or k-mer composition, either with potential host genomes [7,9,10], or with reference virus genomes [11]; and machine learning methods using a variety of sequence derived features as described below. Although the first strategy can give high confidence predictions, the predictions are constrained by the limits of alignment approaches at low sequence similarity. K-mer profile comparison provides alignment free methods but because of lack of contrast when measuring proximity in high dimensional space they lose discriminative power. Additionally, all methods that rely on reference genomes are constrained by the genomes available in the databases. Machine learning approaches offer alternatives that are not dependent on reference genomes or alignment, relying instead on a set of labelled training examples. They are widely used in computational biology to aid in the analysis of large biological sequence datasets [12] due to their ability to find weak patterns in complex and noisy data without requiring prior knowledge of the specific mechanisms responsible for the phenotype of interest [13]. A key factor for successful machine learning is measurable attributes, termed 'features', of the data. These are typically represented in fixed length numerical vectors that encapsulate the discriminative information contained in the data; in this case properties of the viral genomes.

To date, most machine learning approaches to virus host prediction have used features derived from oligo-nucleotide or k-mer biases that are known to correlate with their host genomes, such as: CG bias [14], CpG bias [15–17] and di-codon bias [18]. Di-nucleotide features, in particular, have been included in a wide range of virus host prediction tasks, from training on a single virus species or genera with multiple hosts such as rabies virus, coronavirus, and influenza A virus[19–21], to training on host taxa with multiple viruses [22,23]. The potential for improved prediction by extending the length of the nucleotide k-mers has been demonstrated by Zhang et al. [24]. The nucleotide sequence contains the information needed for a virus to exploit its host: regulatory RNAs, amino acid sequences etc. The latter, through their biochemical properties, fold into three dimensional structures with functional properties mediated through molecular interactions. Although all of this 'functional' information is present in the nucleotide sequence, it is not necessarily in a form that is easy for machine learning approaches to extract. Only two machine learning approaches have previously demonstrated the potential of using alternative representations of the genome for virus host prediction: Raj and co-workers [25] successfully used amino-acid k-mers to predict the host kingdom of two RNA virus families, while Leite and co-workers [26] included predicted domain-domain interactions in their features to predict phage-bacteria interacting pairs.

The use of features derived from viral genomes for host prediction is based on the observation that over time, the coevolutionary virus-host relationship embeds a host specific signal in the virus genome [5,27–30]. As obligate intracellular parasites, viruses must enter a host cell, subvert its defence systems and exploit its cellular systems to replicate [31,32]. To achieve this the virus must make hundreds to thousands of molecular interactions with the host system while evading the host immune response [33]. This antagonistic relationship drives a coevolutionary 'arms race' imprinting a host specific signal in the viral genome. The majority of virus-host interactions are protein-protein interactions known to be mediated through both domain-domain interactions and domain-motif interactions. For example, there are many known examples of viruses converging on host short linear motifs (SLiMs) [31,34,35] to directly mimic host molecular interfaces [36]. Indeed it has been observed that pathogens from across different domains of life that infect the same host, mimic the same host motifs [37].

The phylogenetic signal due to the evolutionary relationship between viruses that infect the same host can also be predictive. This is due to the coevolutionary process resulting in a tendency for virus and host phylogenetic trees to be congruent despite the occurrence of frequent host-switching. Host switching tends to be preferentially biased to closely related hosts [38,39] because, for a virus to successfully jump host species, it must be able to evade the immune response and exploit the new host's system for replication. Babayan et al. [22] successfully made use of this signal by combining a measure of 'phylogenetic neighbourhood' with selected features derived from nucleotide biases to predict the reservoir host of newly emerging viruses. Alignment free phylogenetic analysis [40] has shown that k-mer composition contains sufficient phylogenetic signal to reliably infer evolutionary relationships. Protein domains have also been used to infer phylogeny, for example, Phan et al. [41] recently used domains to classify newly discovered coronaviridae genomes.

The goal of this study was to compare the predictive power of a wider range of features than those generated from nucleotide sequences alone. We hypothesise that features derived from transformations of nucleotides sequences to other representations have the potential to improve prediction over nucleotide sequences as they make the complex nature of both the evolutionary and host-mimicry information more easily accessible to machine learning algorithms. To do this we transformed the genome sequences to higher level sequence representations by adding biological information in the form of: (i) translation of nucleotide to amino

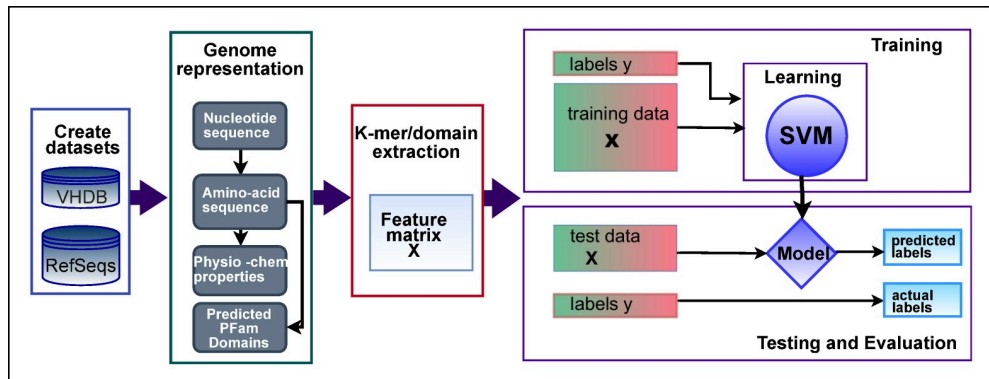

**Fig 1. Workflow for extracting and testing different level feature sets for predicting host taxon information.** Virus genome data was represented by four information layers and features derived from each. Binary classification with linear SVM was used on the equal sized positive and negative classes of virus-host association, split into training and test sets. Area under the ROC curve, AUC, score was measured for each dataset-feature set combination.

acid sequences; (ii) physio-chemical properties of amino acid residues—a more functional representation that allows for conservative amino acid substitution; and (iii) predicted protein domains—distinct structural/functional subunits of a protein that are associated with specific functions.

Based on these four representations of the viral genomes and using a supervised machine learning workflow (Fig 1) we found that all levels of feature representation are predictive of host taxonomic information for both prokaryote and eukaryote hosts. Furthermore, by using a novel phylogenetically-aware 'holdout' method, we were able to investigate the contribution of phylogenetic and convergent signals to prediction. Using a kernel combination method to improve prediction we show that the information embedded by these different genome representations is complementary and can be combined to improve predictions. Our results demonstrate the use of features that capture the different layered biological signals arising from multiple types of the viral-host molecular interactions has the prospect of improving the accuracy of virus host prediction across a broad range of applications.

## Results

Datasets for different host taxa, or labels, were created using sequences for positive and negative viruses, that is, viruses that are either known or not known to infect the labelled taxa. To ameliorate the problems caused by overlapping or redundant data we kept a minimum distance between sequences by using only the reference sequence for each viral species. The Virus Host Database [14] was used to identify known species-level virus host interactions for both prokaryote and eukaryote hosts at different host taxonomic levels. We created a balanced binary dataset for each host taxa for which there were more than a minimum number of known interacting viruses. Known interactions made up the positive labelled class. The negative class was drawn from the remaining viruses that infect hosts in the parent taxa of the positive class. By setting a low threshold of 28 viruses as the minimum class size, giving a total dataset size of 56, we were able to analyse multiple datasets at species and genus level. For the prokaryote hosts this resulted in 65 datasets (all for bacteria hosts—which we refer to these as the 'bacteria datasets'), corresponding to Baltimore class I, dsDNA viruses (S1 Table).

For the eukaryote hosts, this procedure resulted in very few host taxa below class reaching our threshold for minimum dataset size. We therefore used the following two strategies: combining viruses from all Baltimore classes; and combining all RNA viruses, respectively, for a

particular host taxon into a single dataset. This resulted in a total of 116 eukaryote datasets covering 57 host taxa over all taxonomic ranks, from kingdom to species level and the different Baltimore, and combined classes of the viruses (S2 Table). These include 48 datasets comprising all RNA viruses for host groups that include many at family, genus and species level.

Each of the 224 datasets was randomly split into training and test partitions with a ratio of 0.8 to 0.2 prior to extracting the 20 different feature set matrices from the viral genomes (Table 1). Each of these feature matrices was used to train and test an SVM classifier, resulting in AUC scores for over 3740 classifiers.

## Features from all genome representations are predictive of host across all hosts

To test the predictive capacity of the different levels of the genome representation we trained and tested a binary classifier for each of the 20 feature sets on all of the datasets described above. The results of the evaluation of all the classifiers demonstrate that all levels of genome representation contain a signal predictive of host taxa across the host tree. Heatmaps comparing the AUC scores for the prokaryote (Fig 2) and eukaryote (Fig 3) classifiers show that apart from DNA k-mers of length 1, all feature sets are consistently predictive. In particular, omitting results for DNA k-mers of length 1, 82% of the dataset-featureset combinations have an AUC of 0.75 or more (74% with AUC of 0.8 or more) Any AUC score above 0.5 (random classification) indicates the presence of a predictive signal. A score of 1 demonstrates the potential for a perfect classifier where all predictions are correct. Most host taxa have many feature sets that contain a predictive signal (146 out of the 180 datasets have at least one feature set with a score of greater than 0.90). Some hosts are more challenging to predict with none of the feature sets giving good performance, (6 out of the 180 datasets have a maximum score of less than 0.80). This is most apparent at the lower taxonomic ranks of species and genus where we are trying to separate the viruses of more similar hosts and for some Baltimore classes. Overall, the results show that a genomic signature that predicts host taxonomy is present at all levels of biological information representation tested in our study.

## Longer k-mers of all genome representations are more predictive

To compare the effect of k-mer length on prediction accuracy we tested a range of k-mer lengths for the sequence representations of the genomes (nucleic acid, amino acid and physio-chemical properties). The results show that for all feature levels, prediction improves with increasing k-mer length (Figs 4 and 5). This is despite the exponential growth in the size of the

**Table 1. The 20 feature sets generated from the four representations of the viral genomes.**

| Genome representation | [Letter set] (Alphabet size) | K-mers lengths tested (k) | Feature set size for maximum k |
|---|---|---|---|
| DNA | [A,C,G,T] (4) | [1–9] | 262,144 |
| AA | [Amino Acid single letter code] (20) | [1–4] | 160,000 |
| PC | [t-z] (7) | [1–6] | 117,649 |
| **Domains** | [All domains predicted within each dataset] | 1 | Total number of unique domains in all the viral genomes in VHDB—2200 |

Genome representation: DNA—nucleotide sequence; AA—amino acid sequence of CDS regions; PC—physio-chemical properties, each amino acid residue binned into one of seven bins based on its physio-chemical property; Domains—presence of PFAM domain in the sequence.

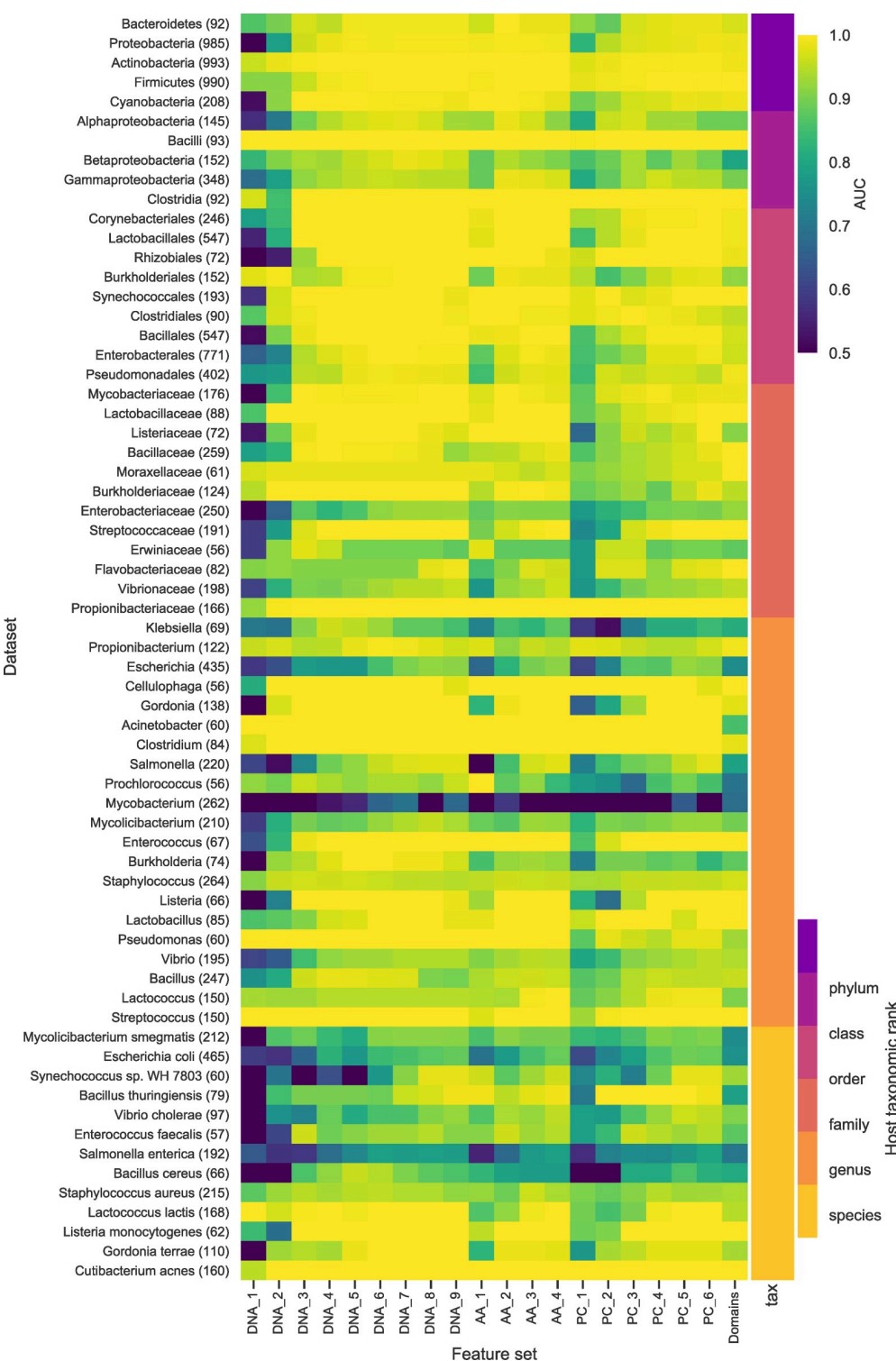

**Fig 2. Comparison of the results for all the bacteria datasets for all the feature sets.** The heatmap shows that all feature sets contain some predictive signal with an AUC > 0.5 for the majority of the bacteria datasets. The rows each correspond to a dataset and are ordered by taxonomic rank (indicated by the colour bar on the right) and each column a feature set. The feature set labels the letters indicate the genome representation and the number the k-mer size. DNA —nucleotide sequence; AA—amino acid sequence of CDS regions; PC—Physio-chemical properties, each amino acid residue binned into one of seven bins based on its physio-chemical property; Domains—presence of PFAM domain in the sequence. The colour indicates the AUC score for each classifier. All AUC scores of less than 0.5 were set 0.5, i.e., no predictive signal. The number of viruses in each dataset is in brackets.

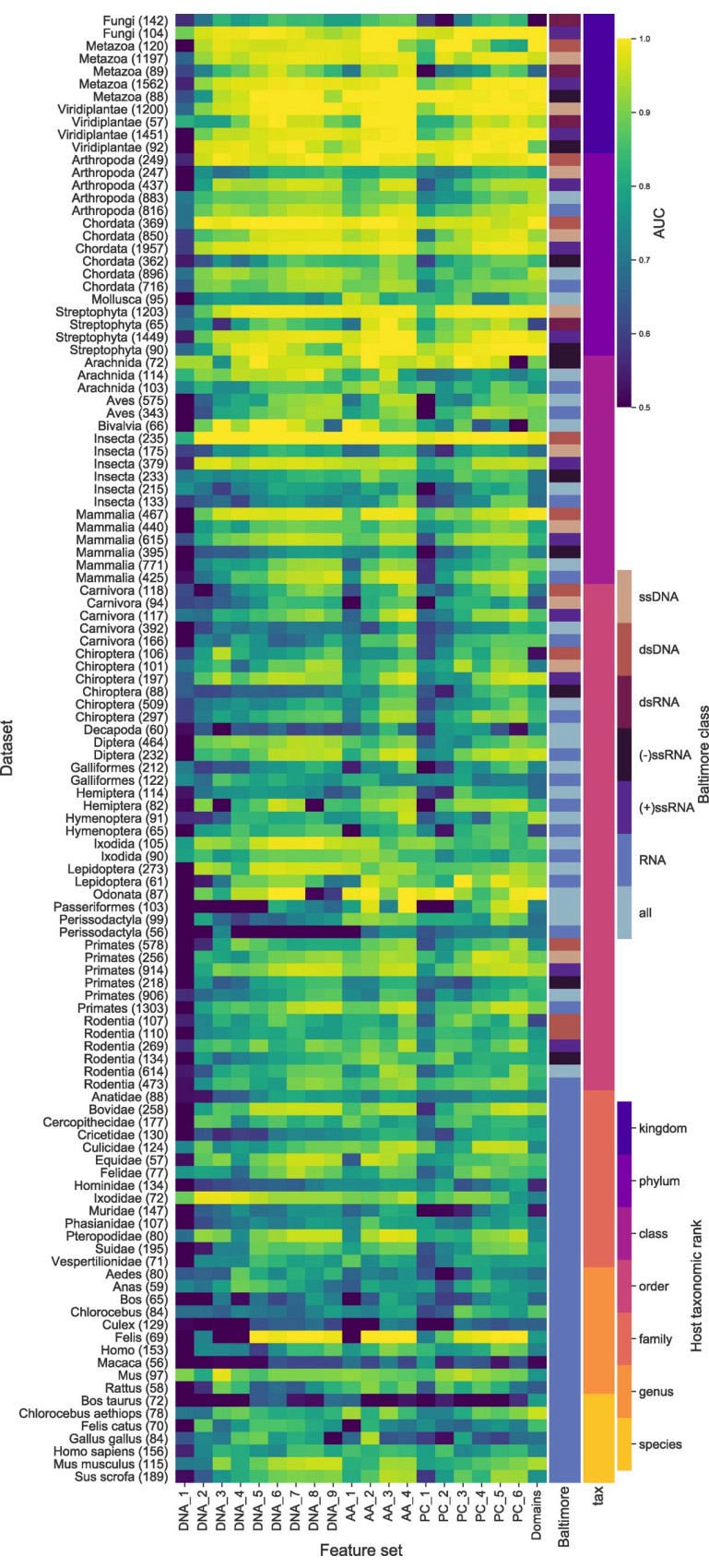

**Fig 3. Comparison of the results for all the eukaryote datasets across all the feature sets.** The heatmap shows that most of the feature sets contain some predictive signal, AUC > 0.5, for the majority of the eukaryote datasets and for all Baltimore groupings (indicated by the inner colour bar on the right). Each row corresponds to a dataset and are ordered by taxonomic rank (indicated by the outer colour bar on the right) and each column corresponds to a feature set. For the feature set labels the letters indicate the genome representation and the number the k-mer size. DNA—nucleotide sequence; AA—amino acid sequence of CDS regions; PC—Physio-chemical properties, each amino acid residue binned into one of seven bins based on its physio-chemical property; Domains—presence of PFAM domain in the sequence. The colour indicates the AUC score for each classifier. All AUC scores of less than 0.5 were set 0.5, i.e., no predictive signal. The number of viruses in each dataset is in brackets.

feature sets, e.g., the DNA_2 feature sets have $4^2$ = 16 possible k-mers, compared to DNA_9, which has $4^9$ = 262144. These larger feature sets are very sparse. For example, the DNA_9 Mammalia dataset for RNA viruses has a sparsity of 0.91, i.e., over 90% of the elements in the sequence by k-mer matrix are zero, although over the 425 virus genomes almost all of the possible k-mers occur in at least one genome.

Prediction appears to be more difficult for the eukaryote datasets (Fig 5). This is perhaps due to the fact that eukaryote hosts are infected by viruses from across all seven Baltimore classes. The alternative replication/life-cycle strategies used by viruses from different classes will involve dissimilar sets of molecular interactions. It is therefore likely that they will acquire disparate host-derived signatures in their genomes, making the classification task more challenging. The problem is further exacerbated by the size of the datasets with few host taxa being available below the rank *class* when split on Baltimore class to meet our minimum dataset size. When testing the datasets formed by combining all RNA viruses, or all Baltimore classes, we were able to widen the range of hosts tested. Although prediction is better when using individual Baltimore classes, there is still a predictive signal when using the combined datasets (Fig 5).

Comparing classifiers for datasets moving from higher to lower taxonomic levels, i.e, from phylum to species level in the host tree, prediction becomes less accurate and less consistent across all the feature sets. For the bacteria datasets at phylum level all of the feature sets with the exception of DNA k = 1, are highly predictive with an average AUC of 0.86 and standard deviation of 0.07 (Fig 4.i). In contrast, the species level classifiers have an average AUC of 0.67 and standard deviation of 0.15 *(Fig 4.vi)*. One possible reason for this drop in predictive power (and increased variance) is the decrease in size of the datasets, as the data is stratified into a larger number of smaller subsets. This is confirmed by comparing the AUC scores against dataset size (Fig 6). Although many of the smaller datasets achieve a high AUC (towards top left of the plot), the worst performing classifiers all correspond to smaller datasets (bottom left of the plot). Finally, many of the worst performing small datasets were generated by including all RNA viruses (denoted by crosses in Fig 6). By their nature, these polyphyletic datasets will likely contain a wider range of host-derived mimicry signals than datasets comprising individual Baltimore classes, and it seems reasonable that they would therefore suffer more from the lack of a large number of training examples.

## The predictive signal contains both phylogenetic and convergent elements

Next we performed experiments to determine if we were finding more than just a phylogenetic signal embedded in the virus genomes. We developed a novel cross validation method where, rather than stratifying data randomly into training and test sets we withheld one complete virus family from training and then used it to test the resulting classifier (Fig 7). Our aim was, as far as possible, to holdout a group of closely related viruses. High predictive performance on the holdout family would imply a strong host-specific signal, i.e., predictive signals can be generalised across viruses for the same host. Poor predictive performance would indicate a signal that was specific to that particular virus family. High predictive performance on the holdout

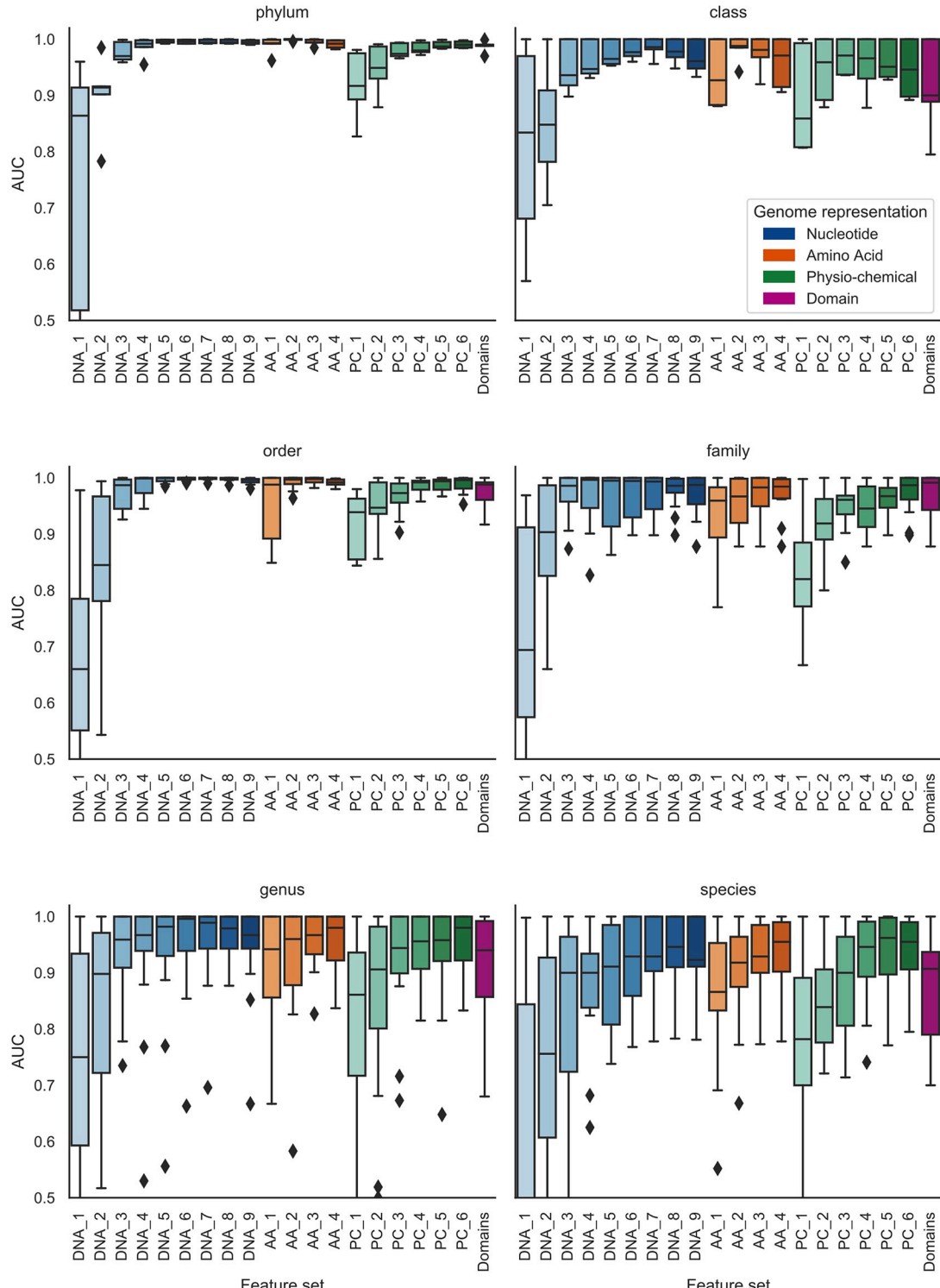

**Fig 4. The effect of k-mer length on prediction across host taxonomic ranks for the bacteria datasets.** The boxplots show how prediction improves with increasing k-mer length for all representations of the genome and that prediction gets more difficult at lower taxonomic ranks. Genome representation is indicated by colour and k-mer length by depth of colour:DNA—nucleotide sequence (blue); AA—amino acid sequence of CDS regions (orange); PC—Physio-chemical properties, each amino acid residue binned into one of seven bins based on its physio-chemical property (green); Domains—presence of PFAM domain in the sequence. Any AUC scores of less than 0.5 were reset to 0.5, i.e., no predictive signal.

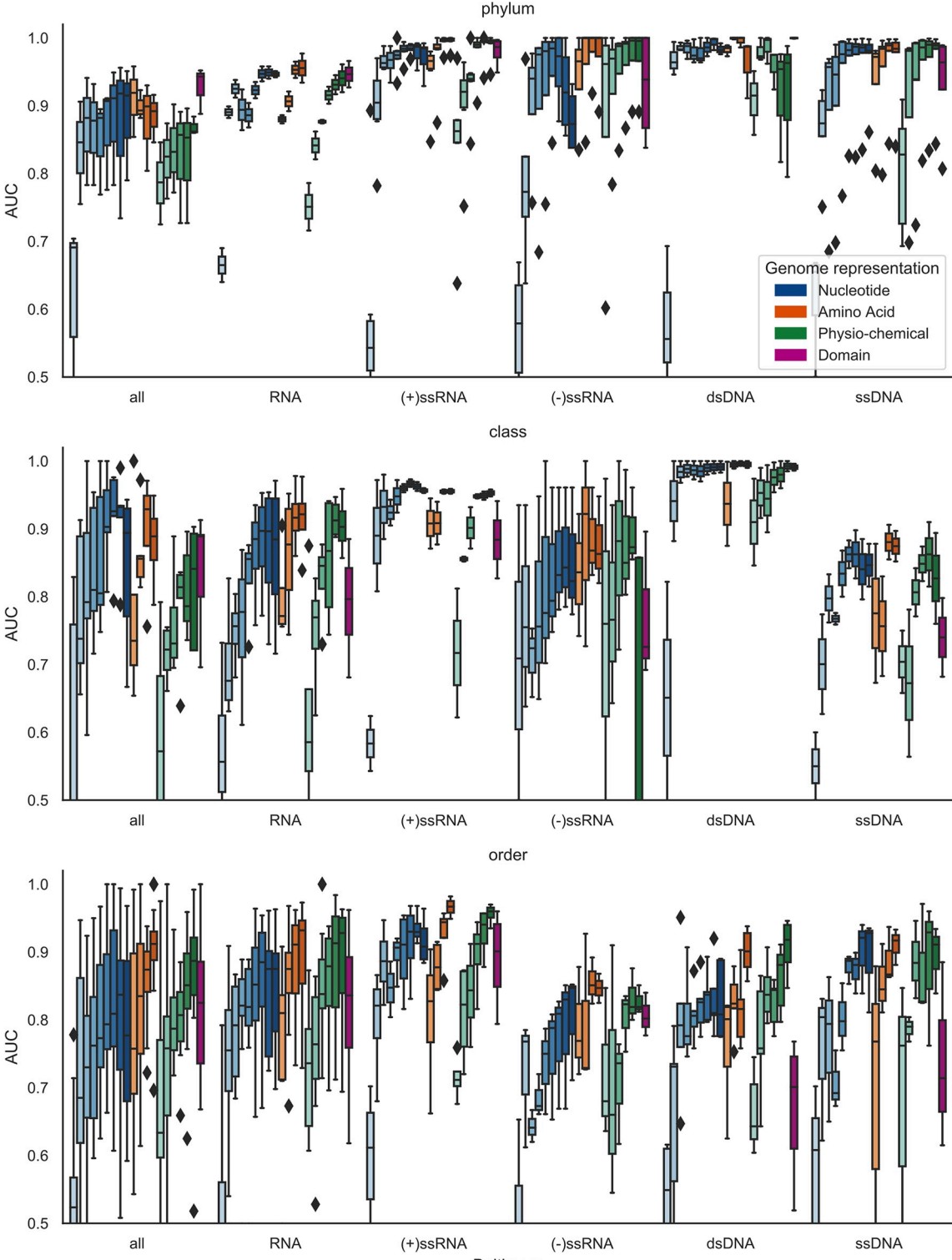

**Fig 5. The effect of k-mer length on prediction across host taxonomic ranks for the eukaryote datasets.** As with Fig 4 we see prediction improves with increasing k-mer length comparing prediction across the different Baltimore groupings. These boxplots show how prediction improves with increasing k-mer length for all representations of the genome and that prediction gets more difficult at lower taxonomic ranks. Genome representation is indicated by colour and k-mer length by depth of colour: DNA— nucleotide sequence (blue); AA—amino acid sequence of CDS regions (orange); PC—physio-chemical properties, each amino acid residue binned into one of seven bins based on its physio-chemical property (green); Domains—presence of PFAM domain in the sequence. Any AUC scores of less than 0.5 were reset to 0.5, i.e., no predictive signal.

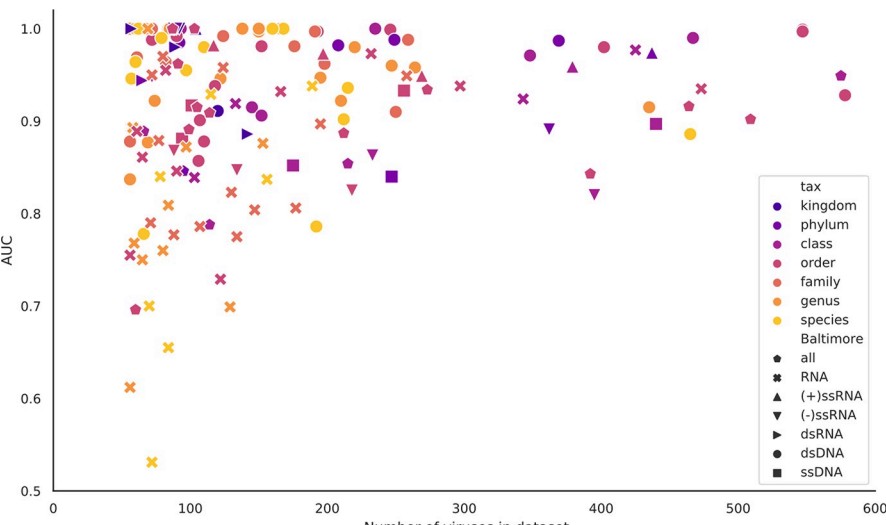

**Fig 6. Comparison of the AUC scores against the size of the datasets.** The scatterplot shows that most of the classifiers achieve good AUC scores (above 0.85). This is the case even for the small datasets and for those at family level and below. The points are coloured by the host taxon level and shaped by Baltimore group. All the classifiers are for AA_4 feature sets. All AUC scores of less than 0.5 were reset to 0.5, i.e., no predictive signal.

family would imply a strong host-specific signal, i.e., predictive signals can be generalised across viruses for the same host. Poor predictive performance would indicate a signal that was specific to that particular virus family.

We used the bacteria phylum to order taxa groups that are infected by the three major *Caudovirales* families (S3 Table). This choice was made to allow us to create datasets with enough viruses of both positive and negative classes, in equal numbers, in both the training and hold-out/test datasets to train and test 'holdout' classifiers. Although the ICTV virus taxonomy for bacteriophage does not reflect phylogeny at family level there is currently no consensus on a method to phylogenetically classify viruses at levels deeper than genus. Using the multiple genera within the families as a group enabled us to select large enough datasets to reasonably train and test a 'holdout' classifier. While phylogenetic based classification systems such as Victor [42] and Gravity [43] do broadly support the ICTV assignments, finding distinct clades at genus level, they also demonstrate that there are inconsistencies in bacteriophage classification with some unrelated viruses in the genera.

Despite holding out a virus family, it could still be the case that there are viruses in the training set of very high similarity to those that had been held out. To remove these, we filtered training sets using average nucleotide identity (ANI) with a threshold of 75% identity across an alignment of at least 10% of the shortest genome. ANI has become established as a reliable and robust method for identifying phylogenetic relationships [44] and recently it has been proposed that pairs of viruses with ANI of greater than 95% over an alignment fraction of 85%, are part of the same species [45]. Because alignment becomes unreliable at lower sequence similarity FastANI has a minimum cutoff of 75%. To ensure that we were removing more distantly related viruses we reduced the alignment fraction to 10%. Our experimental setup is depicted in Fig 7.

To create the holdout datasets we selected bacteria taxa that had multiple Caudovirales families infecting them in large enough numbers to form reasonable sized training and test sets. This requires that the holdout groups and remaining viruses must have a mixture of both positively and negatively labelled viruses (S3 Table). As we were expecting a big loss of signal, we

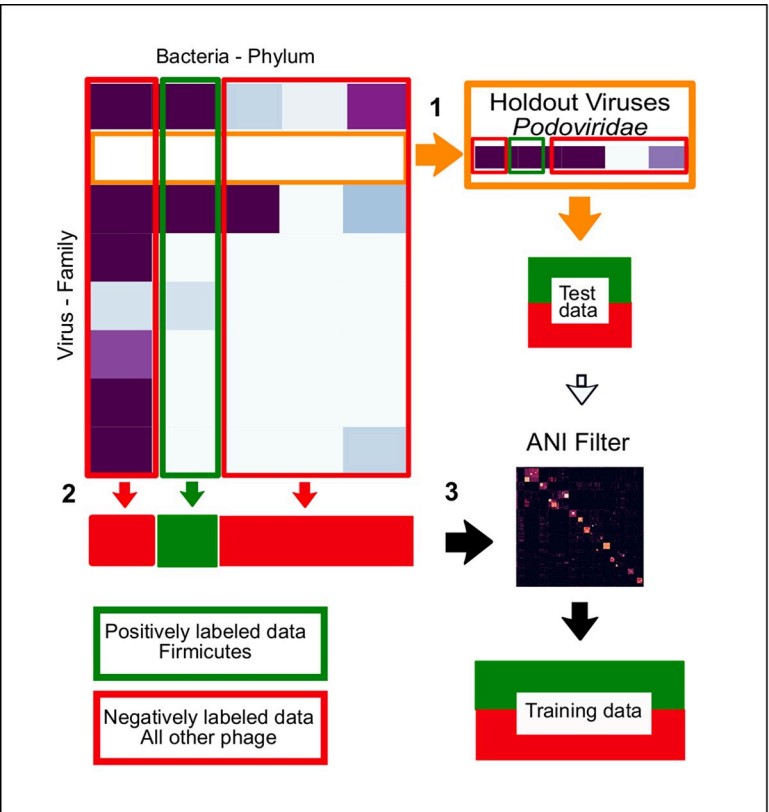

**Fig 7. Creating the holdout datasets.** This shows an example of how a holdout dataset was created. Using the virus host interaction matrix for bacteria hosts at the phylum level and the viruses at family level, the holdout datasets were made by: (1) Removing a family of viruses, here *Podoviridae*, from the data. These holdout viruses are made up of infecting/non-infecting viruses and are then used as the test data. (2) The rest of the viruses that infect/don't infect the labelled host. Here the phylum Firmicutes are used to form the training set. And, (3) The training viruses were then filtered to remove any viruses that have greater than 75% ANI to any of the holdout/test viruses.

chose the most predictive feature sets of each genome representation, along with di-nucleotides to serve as a baseline.

To assess the performance of these 'holdout classifiers', we compared them with our previous classifiers (referred to as 'all'), where a random split of all the viruses was used to form both the training and test sets (Fig 8). Interestingly, while we observed a small drop in AUC performance across all feature sets, we found that the majority of the 'holdout' classifiers retained a predictive signal. The mean ratio between AUC scores of the holdout classifier and standard classifier for the same dataset was 0.86. Across the different feature sets tested this mean ratio ranged from 0.77 (PC_6) to 0.94 (DNA_9) (Fig 9). This small drop in performance demonstrates that a predictive signal is still present, suggesting a common signal that is specific to viruses that infect the labelled host. This indicates there is convergence on a set of host-specific mimicry signals, such as molecular interactions, that is shared across all virus families that infect the host taxa (including the holdout 'family'). In a few cases there is a complete loss of signal, which we hypothesise to mean that the signal learned when training the classifiers on all the viruses includes an element that is specific to the holdout family, and this part of the signal will be absent when training the holdout classifier.

In terms of comparing the different genome representations, it is difficult to identify a consistent pattern as to which feature sets have the biggest signal loss, although protein domains

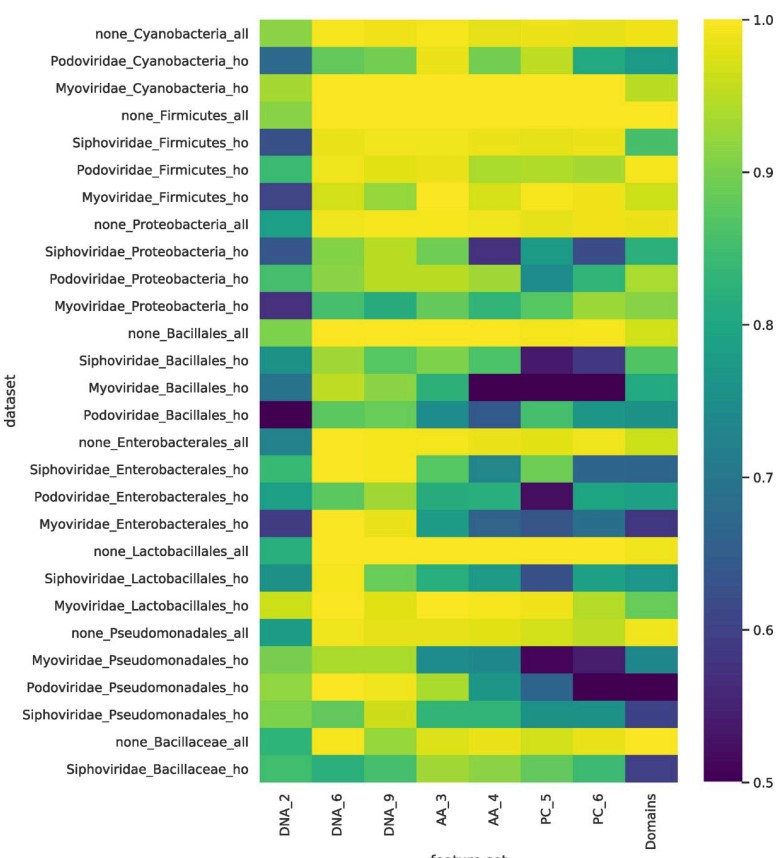

**Fig 8. Comparison of the 'holdout' and 'all' classifiers showing the signal loss.** Comparison of holdout and the standard (labelled 'all') classifiers for each dataset. For the majority of datasets there was a small loss in predictive power, implying that both classifiers are learning a shared signal. In a minority of cases there was a complete loss in predictive power implying the lack of a common signal. Each row corresponds to a dataset and each column a feature set. In the feature set labels the letters indicate the genome representation and the number the k-mer size. Genome representation: DNA—nucleotide sequence; AA—amino acid sequence of CDS regions; PC—physio-chemical properties, each amino acid residue binned into one of seven bins based on its physio-chemical property; Domains—presence of PFAM domain in the sequence. The colour indicates the AUC score for each classifier. All AUC scores of less than 0.5 were set 0.5, i.e., no predictive signal.

(Domains_1), and physio-chemical property derived features (PC_5, PC_6), have more datasets where the holdout classifiers have a big signal loss (Fig 9) than the other representations. Whilst the majority of Domains and PC datasets remain predictive, roughly a quarter of the datasets have a large drop in AUC, with ratios of less than 0.75. As a comparison, none of the DNA_6, DNA_9 and AA_3 datasets had a ratio of less than 0.75. Physio-chemical features are not changed by conservative amino acid substitutions. One possible explanation for the drop in performance of PC features is that as sequences diverge, they will remain more similar at the PC level than at nucleotide and AA levels. Likewise, protein domains remain more identifiably homologous in divergent genomes, whereas convergence of domains is rare [46]. Removing the signal originating from the phylogenetic relationships between viruses in the holdout datasets may therefore lead to a larger drop in AUC for these more evolutionary-linked features. Cases where the domain signal is not lost may indicate a distant phylogenetic relationship or be due to shared domains arising as a consequence of horizontal gene transfer (HGT).

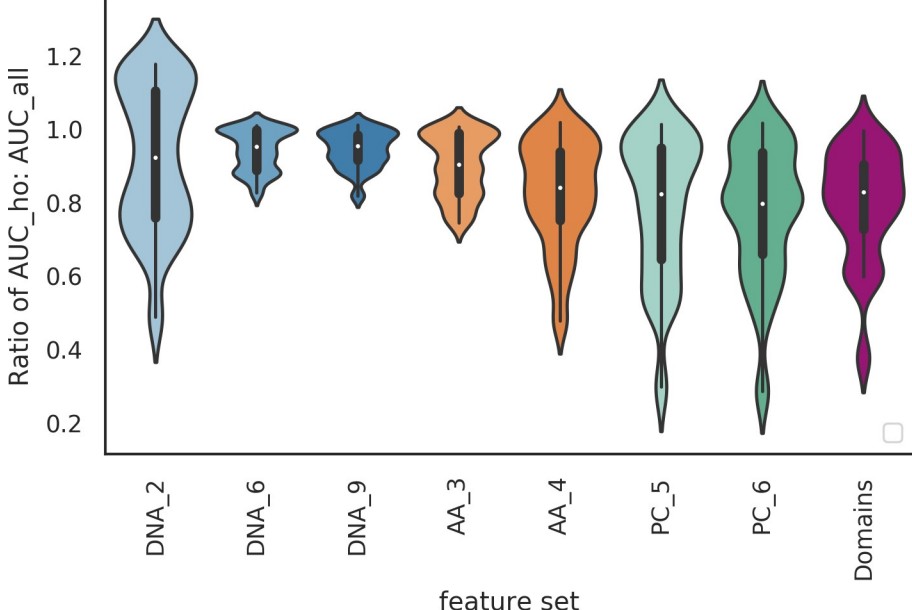

**Fig 9. The signal loss for holdout classifiers.** Violin plots of the ratios of the AUC scores for holdout (AUC_ho) to standard (AUC_all) classifiers for each dataset showing the variation in signal loss for the different feature sets. For the feature set labels, the letters indicate the genome representation and the number the k-mer size. Genome representation: DNA—nucleotide sequence (blue); AA—amino acid sequence of CDS regions (orange); PC—physio-chemical properties, each amino acid residue binned into one of seven bins based on its physio-chemical property (green); Domains—presence of PFAM domain in the sequence.

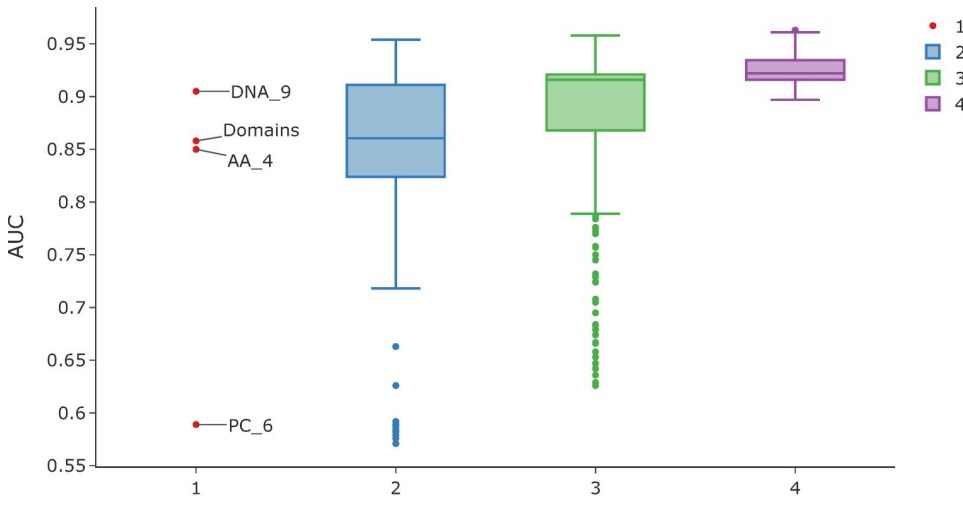

**Fig 10. Combined kernel classifiers.** This shows an example of how prediction can improve with the number of kernels contributing to the SVM classifier. This shows the results for all the iterations for combining kernels grouped by the number of kernels contributing to the combined kernel, for the dataset for the host order *Bacillales* with holdout group *Siphoviridae*. The red points are the results for the single kernels classifiers: DNA_9—nucleotide sequence kmers length 9; AA _4—amino acid kmers of length 4; PC_6—physio-chemical properties of amino acid sequence kmers length 6; Domains—presence of PFAM domain in the sequence.

## Feature sets from the different genome representations contain complementary information

The overall aim of our study was to investigate whether using features extracted from different levels of viral genome representation were predictive of host. To check whether these alternative features are redundant or provide complementary information we combined feature sets from the different genome levels. A property of kernels, as used by SVMs, is the fact that it is straightforward to combine feature sets by creating composite kernels[47]. We thus combined the most predictive kernels from the DNA_9, AA_4, PC_5 and Domain feature sets, in different linear combinations. Weights for each kernel were varied between 0 and 1 (in steps of 0.05) with the sum of the weights across the four kernels constrained to equal to 1. To test if this improved prediction, we selected a poorly predicted dataset as an example. We used the holdout classifier for the host label *Bacillales* and holdout group *Siphoviridae*. The results for the single kernel classifiers were DNA_9 = 0.91, AA_4 = 0.85, PC_6 = 0.59, Domains = 0.86. A summary showing the results for each classifier, grouped by the number of kernels contributing to the combined kernel, is shown in Fig 10. This demonstrates that overall, prediction improves as more kernels, drawn from the different genome representations, are included in the combined kernel. Furthermore, using different kernels could be used as a method to tune the classifier on the metric of importance. For example, Fig 11 shows the same data as the false discovery rate (FDR) against true positive rate (TPR) for each classifier. Conversely, when different k-mer lengths from the same genome representation were combined, no improvement in prediction was seen.

## Discussion

The aim of this study was to compare the predictive power of a wide range of features for use in machine learning approaches to virus host prediction. We generated 20 feature sets from multiple representations of viral genomes and tested their capacity for host prediction. We found that features derived from all representations are predictive of host taxon for both bacteria and eukaryote hosts (Figs 2, 3, 4 and 5), and that different features contain complementary signals that can be combined to improve prediction (Figs 10 and 11). Through a phylogenetically aware stratification scheme (Fig 7), our results strongly suggest that the features capture both phylogenetic and convergent signals (Figs 8 and 9).

The majority of previous machine learning approaches to virus-host prediction have focused on information from nucleotide sequences only [19–21,23,24], which although predictive of host, ignore the rich information contained within alternative representations of the genomes. Through a process of convergent evolution, viruses are known to mimic their host's molecular interfaces at domain-domain and domain-motif interaction sites [34,48,49]. Such mimicry will be reflected in the amino acid sequence and domain content. Our results show that features derived from these genome representations can be successfully used for prediction, as demonstrated in previous studies [25,26].

Although we see evidence of a predictive signal in all of the representations across the host tree, there is no universal best feature set. In addition, some datasets are more challenging to predict with none of the feature sets achieving good performance. This is most apparent at the lower taxonomic ranks of species and genus where we are trying to separate the viruses of more similar hosts. Some Baltimore classes are easier to predict than others. For example, classifiers for both bacteria and eukaryotic dsDNA viruses consistently achieve higher AUC scores than RNA viruses, presumably because the prevalence of HGT means that similar sequences occur in viruses with a shared host. Conversely all of the eukaryote RNA datasets will be affected by their fast mutation rates leading to loss of sequence similarity.

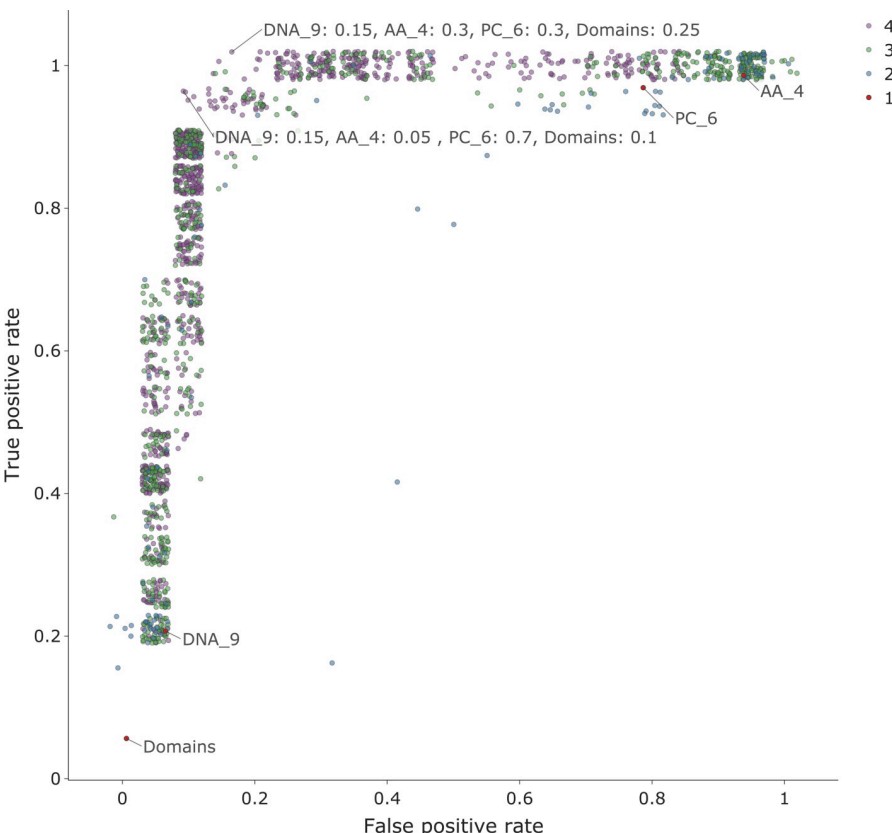

**Fig 11. A plot of false positive rate (FPR) versus true positive rate (TPR) for the combined kernels of one dataset.**
By adjusting the contribution of the different kernels, we can alter the specificity (1- FPR) and sensitivity (TPR) of the classifier. Each point represents the results for a classifier, each with a different combination of kernel weights, with the number of kernels shown by the point colour. The red (labelled) points are the results for the original single kernel classifiers. Additionally, two of the best classifiers have been labelled with the kernel contributions. This shows the results for all the iterations for combining kernels for the dataset for the host order *Bacillales* with holdout group *Siphoviridae*. The data points have been 'jittered' to reduce the overlap. The kernels used were: DNA_9—nucleotide sequence kmers length 9; AA _4—amino acid kmers of length 4; PC_6—physio-chemical properties of amino acid sequence kmers length 6; Domains—presence of PFAM domain in the sequence.

Our novel holdout method suggests that the predictive signal embedded in viral genomes is made up of both phylogenetic and convergent signals. We removed, as far as possible, the signal coming from the phylogenetic relationships between the viruses infecting a host and found that the majority of our 'holdout' classifiers still contained a predictive signal. Possibly the signal remaining is due to the convergence of both the training and 'holdout' viruses on common host molecular interfaces or other host factors. Hence, we hypothesise that the predictive signals in the viral genomes are a combination of two elements: the phylogenetic or virus-specific element which is removed from training causing the loss in prediction; and the convergent or host-specific element which remains still allowing for some prediction. By contrast, 'holdout' classifiers that have a complete loss in signal, indicate that there is no convergence and that the 'holdout' viruses are probably using different molecular interactions than the 'training' viruses.

Our results show that increasing the length of the k-mers improves prediction with all sequence representations. Although many machine learning approaches have used di-nucleotide features[19–23], other computational approaches have shown that using longer k-mers (length 6 and 8) is beneficial to prediction [9–11]. Zhang [24] found that with random forest

classification, increasing nucleotide k-mers up to a length of 8 improved prediction. Interestingly, even though we might expect long k-mers to perform badly due to mismatches, we found that for the ranges of k-mer lengths we tested, prediction increases with length. This is in accordance with HostPhinder[11] that successfully used co-occurring k-mers of length 16 to predict hosts and the finding that, even after controlling for HGT, much longer nucleotide sequences co-occur in viruses and their host across all classes of viruses and host [50].

Longer k-mer features and domains—which only occur once or a few times in a genome—have the capacity to encode information about local virus-host molecular interactions, such as motif-domain or domain-domain interfaces. This is opposed to the shorter oligo-nucleotides which occur multiple times in a genome and therefore gives a global measure of biases over the whole genome. Changes in the occurrence of these local features caused by single mutations as a virus adapts to its host, will have a big impact on k-mer composition, whereas global genome-wide biases will take many mutations over the whole genome to have a significant effect on the k-mer composition. This will result in low dimensional biases being slower to match their host's bias, broadly agreeing with the findings of Di Giallonardo et al. [51] that dinucleotide composition of a virus is more closely related to its family taxa than its host species.

Machine learning requires suitable training examples. We are therefore constrained to making predictions about the small fraction of cellular life that have many known viruses. This data is biased towards well studied organisms such as humans or pathogenic bacteria. To overcome this, we have pooled hosts into higher taxa. For all feature sets, prediction gets more difficult for datasets at lower taxonomic ranks, with AUC scores being consistently higher for the phylum level datasets than those at species level. There are three possible causes for this deterioration. Firstly, the volume of data available for training decreases, although we see that all the worst scoring classifiers are for smaller dataset, many of the smallest datasets also achieve high scores. Secondly, the negative data are viruses that are not known to interact with the host and may include viruses for which interactions have not yet been observed, i.e., there may well be false negatives in our training/testing sets which can result in predictions incorrectly labelled as false positives. In addition, because viruses tend to infect closely related hosts this mislabelling will be more likely to occur at lower taxonomic ranks. Finally, as we move from higher to lower host taxonomic ranks, we are trying to discriminate between the viruses of more similar hosts—a more challenging problem. The lack of correlation between the size of the dataset and score is very apparent at genus and species level, (Fig 2), with the classifiers for the datasets *S. enterica* (192 viruses), *E.coli* (465 viruses) and the genus mycobacterium (262 viruses) performing particularly badly across all feature sets. This poor performance may be due to the fact that these groups of viruses are highly diverse and mosaic [52] or that they contain a high number of viruses that infect multiple hosts confounding any host specific signal. All these factors limit the specificity of our predictions and as virus host interactions tend to be species specific, or for bacteria, strain specific, this will limit the applications of this approach. While we restricted our study to using species reference sequences, a wider study using all available host labelled data from databases such as MVP database [53] or the NCBI Virus genome resource [54] should enable higher resolution predictions.

In this study, we have limited the sequence composition derived features (nucleic acid, amino acid and physio-chemical properties) to fixed k-mers, not allowing mismatches. This is a rigid representation of the sequences where information from closely related k-mers, such as those differing by a single mutation, is lost. Most functional elements in biology are better represented by motifs, where some positions in the short sequence are crucial to function and are conserved while others are more variable. Using a motif representation or relaxed k-mers with mismatches, such as used by Raj [25] may be better at generalising across closely related subsequences and ultimately improve performance.

Future development and deployment of classifiers for different virus host prediction domains would require task dependent optimisation of the models, and their operating thresholds. Various model optimisations are possible, including combining multiple feature sets. Our results show the potential of combining sets of features from different genome representations but that there is no consistent pattern as to which feature set works best for different classification tasks. Along with other model parameters, kernel weights would need to be optimized with respect to the most important error metric for the task in hand (Fig 11). For example, in environmental metagenomics minimising type 1 errors (the false discovery rate) is most important. Conversely, when trying to identify the reservoir source of a spillover virus, reducing type 2 errors is critical. This optimisation should include some measure of the prevalence of the data in order to take account of class imbalances. For specific tasks it may also be important to test the effects of partial or incomplete genomes on the performance of the classifiers to ascertain the usefulness and robustness of these features for use in metagenomics. Due to the extensive nature of this study, with over 3500 classifiers trained and tested, we did not perform any optimisation steps in our machine learning workflow. We would expect significant improvement in prediction as a result of using an optimisation process that involves both feature selection/combination and hyperparameter optimisation such as multi-kernel learning [47].

In conclusion, our results demonstrate that features derived from all four representations of viral genomes are predictive across the host tree. Combining a broader range of features that encapsulate the multiple layers of information held within viral genomes can lead to improved accuracy of virus-host prediction. This use of complementary features will lead to higher confidence assignments about host taxon information for the increasing numbers of viruses with unassigned hosts from metagenomics studies and, for example, to identify the reservoir source of a spillover event. Furthermore, the local nature of domain and longer k-mer features have the potential to be informative of the mechanisms leading to virus host specificity.

## Methods

### Data

We downloaded the Virus Host Database (https://www.genome.jp/virushostdb/) [14] on 25/1/2019. The VHDB is a curated database of reported taxonomic interactions between viruses and their known hosts. It is regularly updated from Refseq/GenBank, Uniprot and Viralzone and includes manual annotations. The dataset included 9199 unique viruses associated with 3006 hosts and a total of 14229 interactions. The FASTA files of the reference genome sequences and the amino acid sequences of the coding regions for each virus are also included in the VHDB resources.

### Generating binary datasets from the known virus host interactions

A host taxonomic tree was constructed from all the hosts in VHDB using ETE 3 [55] at the ranks of kingdom, phylum, class, order, family, genus and species. Each host node was annotated with the viruses known to infect it. The tree was 'pruned' to include only nodes infected by at least a minimum number of virus species. The minimum number of infecting viruses was set to 28 for a positive node. As we were comparing how predictive these feature sets were across all taxon ranks, setting this arbitrary threshold at 28 enabled us to include more examples of genus and species level datasets.

For each binary dataset the positive class consisted of the viruses that infect a host node, while the negative class contained an equal number of viruses selected from those that infect the rest of the hosts in its parent node, i.e., the node's most closely related hosts. For example,

for primates the order *primates* made up the positive class, the viruses to form the negative class were selected from those that infect at least one host in the rest of the taxon class *mammalia* (Fig 12). In cases where the negative class comprised fewer than 28 viruses the class was widened to include the next taxa up until at least 28 viruses are present. This resulted in binary datasets of equal numbers of positive and negative viruses with a minimum dataset size of 56 viruses. These were then split into training and test data at a ratio of 0.8 to 0.2 respectively (or 0.75 to 0.25 where the total number of viruses in the dataset was less than 100).

## Genome representation

Four different representations of the genome were used:

1. **Nucleic acid sequence** k-mers (DNA_k) and their reverse complements were extracted from the raw genome sequence in the FASTA files. Segmented viruses were concatenated and treated as a single genome.

2. **Amino acid sequence** k-mers (AA_k) were extracted from the amino acid sequence of the CDS regions in the downloaded FAA files, (segmented viruses were again concatenated).

3. **Physio-chemical properties** of the amino acid sequence, k-mers (PC_k) were extracted by first binning each amino acid into one of seven groups defined by their physio-chemical properties: ({AGV}, {C}, {FILP}, {MSTY}, {HNQW}, {DE}, and {KR}) [56]. The k-mers were then extracted using the seven bin labels as the alphabet.

4. **Domains,** the domain content for each genome was identified using the HMMER package (version HMM 3.2) [57,58]. First a Pfam domain profile database was built using Pfam-A (release 31.0.), then all the amino acid sequences from each genome were scanned against this using the hmmscan command resulting in a list of predicted domains for each virus genome. For each genome in a dataset the domain composition vector was generated by counting the frequency of the unique domains in each virus across the set of all unique domains found within that dataset. These vectors were then normalised to sum to 1. The Hmmscan setting:—cut_tc option was used, this uses the trusted bit score thresholds from the model. The aim being to include the maximum number of possible domains with the expectation that the machine learning will be able to find the true signal above the noise.

## K-mer extraction

The DNA_k, AA_k and PC_k sequence data were represented as a vector of k-mer composition. These were generated by counting the number of times each possible k-mer occurs within the sequence (and the reverse complement of the nucleotide sequences), and then normalising the resulting vector to sum to 1 to account for varying genome lengths. Zhang et al. (2017) showed that this simple method of representing k-mer composition was as effective for predicting host as more complex representations that included background models of the sequence [24]. The maximum length of k-mers at each genome representation level was chosen to restrict the feature set size and keep the workflow computationally reasonable.

## Supervised classification

Linear support vector machine classifiers (SVM) were used for classification. The SVM is well suited to binary classification problems where the number of data points is much smaller than the number of features used [59]. It is also able to cope well with sparse feature matrices where

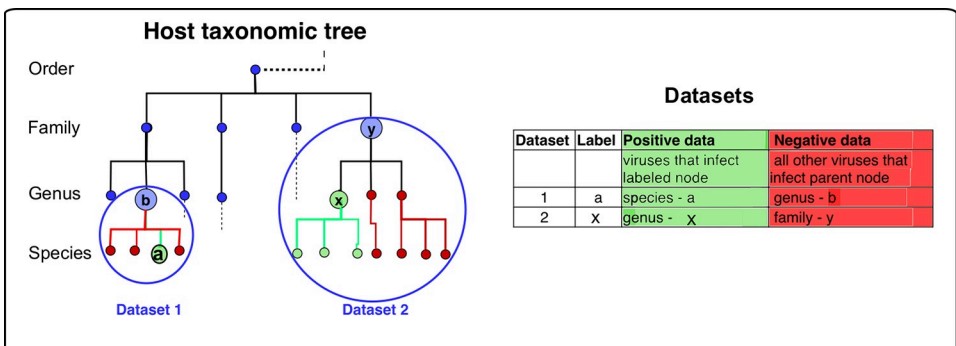

**Fig 12. Generating datasets from the host taxonomic tree.** Datasets were generated from a taxonomic tree of all the hosts with more than 28 known infecting virus species. For each node the positive class consisted of the viruses that infect the labelled node, while the negative viruses were selected from those that infected the rest of the taxon group of that node, for example, if the genus x made up the positive class, the viruses to form the negative class were selected from those that infect the rest of the genera in family y.

many of the elements are zero. Classifiers were trained with the training data from each dataset and then tested for the predictive power of features on the unseen test data. No optimisation steps were performed and a linear kernel was chosen for speed. The python library, Scikit-learn [60], was used with a pipeline that included feature scaling with Standard Scaler. The penalty parameter, C, was left at its default value of 1.0.

Classifier performance was measured using the area under the ROC curve (AUC) score. AUC is a suitable metric for binary classifiers, giving a measure of both specificity and sensitivity without the need to set a threshold. Using a single metric makes it possible to compare the predictive power of the features across the large number of classifiers we tested. The specificity and sensitivity for each classifier are included in the supplementary tables of the results (S4–S6 Tables).

## Creating 'holdout' datasets

To separate the virus and host specific signals our aim was to remove the phylogenetic signal coming from the most closely related viruses from the training data. Holdout datasets were created by first removing a 'related' group (or groups) of viruses from training data and then using these holdout viruses as the test data (Fig 7). Because the ICTV taxonomy does not capture phylogenetic family level relationships, a further step was taken to remove any viruses from the training dataset. Viruses that had more than 75% ANI to any of the holdout viruses were removed from the training data. An ANI matrix of all against all phage was generated using FastANI [44], this matrix was then used to filter viruses that were greater than 75% ANI over a 10% alignment fraction to any of the 'holdout' viruses. FastANI was used with a min-Frac = 0.1 (minimum alignment fraction of genomes) reduced from default of 0.2 as viruses share few common genes and fragment length of 1000, by reducing this from the default of 3000 we aimed to increase the number of shared fragments found.

Again, both the training and test/holdout data contained equal numbers of positive and negative viruses that infected hosts at bacterial host taxa at phylum and order level that had interactions with multiple virus groups at family level. These 'holdout' classifiers were then compared with the standard 'all' classifiers—generated as described above—using a subset of the most predictive feature sets. DNA: k = [2,6,9], AA: k = [3,4], PS: k = [5,6] and domains, (DNA_2 was included for comparison).

## Kernel combination

The goal of this study was to investigate whether using features extracted from different levels of viral genome representation were predictive of host. To check whether these features are redundant we combined the features from the different genome levels. As an SVM is a kernel method we can combine the kernels from the different classifiers in linear combinations to generate a new kernel[47]. We tested a range of kernels on a single dataset by using different weights.

$$K = \sum_{i=1}^{4} w_i k_i \qquad where \qquad \sum_{i=1}^{4} w_i = 1$$

Where $k_1$ to $k_4$ are kernels derived from each genome representation: DNA_9, AA_4, PC_5 and Domains respectively.

All code and data are available at: https://github.com/youngfran/virus_host_predict.git

## Supporting information

**S1 Table. Table of the bacteria host datasets.**
(CSV)

**S2 Table. Table of the eukaryote host datasets.**
(CSV)

**S3 Table. Table of the Holdout datasets.**
(CSV)

**S4 Table. Table of results for the bacteria host datasets.**
(CSV)

**S5 Table. Table of results for the eukaryote host datasets.**
(CSV)

**S6 Table. Table of results for the holdout dataset.**
(CSV)

## Author Contributions

**Conceptualization:** Francesca Young, Simon Rogers, David L. Robertson.

**Formal analysis:** Francesca Young.

**Investigation:** Francesca Young.

**Software:** Francesca Young.

**Supervision:** Simon Rogers, David L. Robertson.

**Visualization:** Francesca Young.

**Writing – original draft:** Francesca Young.

**Writing – review & editing:** Francesca Young, Simon Rogers, David L. Robertson.

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
