## [Decision Letter · Decision Letter 0]

28 Oct 2019

Dear Dr Robertson,

Thank you very much for submitting your manuscript 'Predicting host taxonomic information from viral genomes: a comparison of feature representations' for review by PLOS Computational Biology. Your manuscript has been fully evaluated by the PLOS Computational Biology editorial team and in this case also by independent peer reviewers. The reviewers appreciated the attention to an important problem, but raised some substantial concerns about the manuscript as it currently stands. While your manuscript cannot be accepted in its present form, we are willing to consider a revised version in which the issues raised by the reviewers have been adequately addressed. We cannot, of course, promise publication at that time.

Sincerely,

Morgan Langille, PhD

Associate Editor

PLOS Computational Biology

Virginia Pitzer

Deputy Editor

PLOS Computational Biology

[LINK]

Reviewer's Responses to Questions

**Comments to the Authors:**

Reviewer #1: This study compares of a range of sequence-based signals for virus-host prediction. While the analysis potentially has merit, I have several comments.

Some parts of the text are unclear and should be rewritten/explained better:

- line 67-69

- line 134

- It is unclear what the 120 datasets are (line 143). It might be good to present a Supplementary Table listing all these datasets with the number of viruses.

- line 147: "extracting the 20 different feature set matrices" - which 20? That is unclear at this point and might be presented in a Table.

- In the family holdout analysis it is unclear which families are held out.

- line 267-268

- "suggesting a host-specific signal" (line 270-271): explain better, please don't cut corners

- line 282-284: unclear why would this decrease the ratio?

- I don't follow the argument in line 317-319. Where did you look at synonymous mutations?

- It is unclear what they mean by "precise local virus-host molecular interactions" (e.g. line 321)

- what does "This" refer to in line 331?

- unclear why "these false negatives will occur more frequently at higher taxonomic ranks" (line 337)

- line 411-412

I think the authors may be new to the field as there are several inaccuracies throughout the text:

- line 51 & 361 "orphan genes" are viruses that are not associated with a disease but may possess pathogenicity (wikipedia)

- ref 4 is not the ref for IMG/VR, which should be Paez-Espino et al. 2016 NAR 45:D457 or Paez-Espino et al. 2018 NAR 47:D678.

- viral tagging (ref 5) is not a "low throughput experimental method". It is also unclear how "reliable" it is (line 53).

- The study may benefit from collaboration with a taxonomist, as some terms are confused. For example, the authors consistently mix up high and low taxonomic ranks, and "taxon group" is unclear in line 381 (this should probably be "parent"?).

- They repeatedly set up a straw man that only short nucleotide k-mers have been used until now, e.g. lines 64, 297, 313. This is annoying and does not do justice to a lot of literature, for example see refs 29, 30 (and others) and Edwards et al. 2016 FEMS Microbiol Rev 40:258.

The overlap between training and testing data is not sufficiently discussed. This also affects many of the previous assessments. While the family holdout analysis addresses this to some extent, it is not discussed anywhere how this affects host predictions for new viruses from metagenomes that have not been seen before.

Related to the point above, it is worrying that longer k-mers perform better and unsatisfying that they stopped at k=9. This points to overlap between training and testing data (redundant viruses).

While the family holdout isolates the "host specific signal" to some extent, they did not isolate the "phylogenetic signal" and can thus not conclusively say that the difference in prediction performance fully represents this signal. This should at least be discussed.

The analysis uses only complete viral genome sequences, but metagenomics often yields incomplete genomes. It should be discussed what is the expected effect of this.

They mention that "One reason for this drop in predictive power and increased variance is the decrease in size of the datasets" (line 213, 335). This can be easily tested by selecting the same amount of training data at all levels.

While they controlled for "a minimum number of virus species" (line 376) the diversity of these viruses is not taken into account

I can't find anywhere which "multiple" families were held out in the family holdout analysis (line 246).

Could the "genetic signature that are [IS] shared across all virus families including the holdout ‘family’" (line 272) be due to HGT?

You did not "limited this study to using kmer composition" (line 343) as you also looked at domains.

It is unclear why 28 was the minimum size of the positive set. It is also unclear where the 56 comes from (line 377), but I am ssuming this is 28 positives + 28 negatives. It is also unclear how the total number of viruses in the dataset could be "less than 50" (line 397) of the minimum number is 56.

If the above is correct, this means that the P:N ratio was 1:1 which is not realistic. In nature we expect a huge class imbalance. This should be addressed and/or discussed.

It is unclear how the k-mers were extracted. The method should screen both the leading and lagging nucleotide strands and/or reverse complement k-mers should be accounted for.

Concatenating segments leads to false k-mers overlapping the concatenation site (line 400). This should be avoided.

Settings ad cutoffs of hmmscan are not mentioned. It is unclear how many domains were found.

Figures:

- In the heatmaps it might be insightful to indicate next to the rows how large the datasets were.

- In Figure 3 what are the rows?

- It might be good to provide a Supplementary Table with the actual numbers and refer to it in the fig legend

- Figure 6 is confusing: there are red, green and orange boxes, the word "holdout" is mentioned twice, and there seems to be overlap between training and testing. There is no scale for the purple shading.

- Figure 7A is turned 90 degrees relative to the others. It is difficult to make sense of the order of rows (colunms).

- What is the arrow in Figure 8?

The manuscript is very sloppily written and the authors might consider professional editing. Just some of many examples:

- "keep up with the rapid pace with viral discovery" (line 54)

- words like "literally" and "actually" do not belong in scientific text

- repeated words (line 170)

- mistakes with interpunction (line 186)

- They say an AUC of 0.92 "compares with" an AUC of 0.64 (line 211-213) but this is a huge difference. It is unclear to me where those numbers came from as no figure is referenced.

- Present/past tense mixup (line 249)

- Inconsistencies (Fig. vs Figure, kmer vs k-mer)

- long and convoluted sentences (line 276)

- sentence structure (line 329)

- Supplementary X (line 406)

- Domains -The (line 407)

- They cite the user manual of HMMER.

Reviewer #2: In this manuscript, the authors aim to study the “host” information embedded into viral genomes. To that end, the authors explore and describe the predictive power of different genome features across viral taxonomy, using AUC of a host prediction at individual taxonomic levels to do evaluate each feature.

Overall, the data presented are potentially promising, however in my opinion they are too limited at this point to really inform biology researchers. First, the authors use AUC throughout the manuscript to evaluate the predictive power of individual features, however this metric is, in my opinion, not very informative for a researcher who would need to decide whether or not they should use these approaches in their own research. Instead, sensitivity analyses would be (I think) more useful, including some notions of how many genomes a user could expect to see a prediction for at a given accuracy, and how many hosts would remain ‘unpredicted’. From a tool development perspective, one would need to understand how redundant the different features are by testing different combinations, which is not done in the current study. Hence, at this point, I feel this manuscript describes a potentially interesting set of features that may be used for host prediction, but doesn’t provide enough data for a researcher to use these in their own project, or for tool developers to easily transform these into an automatic host predictor. Finally, and unfortunately, the authors also do not interpret their results from a biology standpoint beyond general descriptions such as “Some Baltimore classes are more difficult to predict than others”, while more detailed interpretations would have made the paper more interesting to a broader readership.

I also noted some problems in the references cited, which are notably missing important tools such as PMID 27899557, 28957499, or 27153081. Given how similar they are to the approach proposed by the authors, these tools should certainly be discussed and ideally compared to the methods presented here (although the last point may be difficult since the author do not really provide a tool per se).

Detailed comments

l. 11-13: This seems to be inexact, VirHostMatcher recommends the use of 6-mers, and WIsH uses 8-mers. It is true though that these tools tend to use only one feature (e.g. nucleotide k-mer) instead of the multi-features approach proposed by the authors, and that’s probably what should be highlighted here instead.

l. 24: “between infecting viruses” is a bit unclear to me, do the authors means “between viruses infecting similar hosts” ?

l. 27: ‘orphan’ could maybe be changed to ‘uncultivated’ which would more specifically defined the scope of the approach proposed here ?

l. 52: Reference [4] seems to be incorrect, as it predates the rise of metagenomic sequence in IMG/VR. Perhaps the authors wanted to reference PMID 31120025 instead, which does reference the IMG/VR database and ~ 5% of sequences with predicted hosts ?

Figure 1: I am a bit confused by the “Feature Extraction” step in this schematic. It only mentions “K-mer composition matrix”, however the authors indicate “Physio-chem properties” and “Predicted PFAM domains” in the Genome representation step before ?

Figure 2: The authors must provide a key for the x-axis abbreviations in the legend, so that a reader can interpret the heatmap.

Figure 3: This heatmap is currently missing y-axis labels (in my version), so that it is impossible to interpret in its current form.

l. 159-160: “some hosts are more challenging to predict”: could the authors maybe expand a little more on which host taxa are more difficult to predict, and what could be the cause of this ? I am especially puzzled by mycobacterium and synechococcus which seem to be associated with the lowest AUC, yet would also be in my opinion the only two groups for which we have enough phage genomes to robustly evaluate the authors’ approach.

l. 184: “for, example,” should be “, for example,”

l. 228: Section “The predictive signal contains both phylogenetic and convergent elements.”

This section is interesting in its attempt at deciphering whether the classifier is trained on a phylogenetic signal (i.e. viruses of the same host look like each other) or some other / distinct host adaptation features. However, I am worried that the approach used by the authors (leaving out a family) is too coarse to be really informative, given the massive variation of phylogenetic similarity between genomes across families (as currently defined). There is also a number of misclassified genomes that could easily lead to a fake impression of a “convergent” signal while the classifier would really be based on virus-virus similarity. If the authors mean to investigate this aspect robustly, they should e.g. perform all-vs-all whole genome ANI for phages, and then hold out based on these ANI values (i.e. not including any genome pairs with ANI > XX% in the training vs test set).

l. 240: “Podoviridae” and all other viral taxon names should be italicized

l. 265: should “lose” be “loss” in “signal lose” ?

l. 299: “ignoring” should be “ignore”

l. 406: “Table Supplementary X” should be replaced with one of the supplementary tables

Reviewer #3: The work describes the use of SVM classifiers to compare different feature sets for their predictive power in viral host prediction. The massive increase in virome datasets makes such predictive tools an extremely important asset in the analysis of such data. The work described shows how SVM can be used to predict viral hosts. The manuscript is generally well written, providing an introduction to the research question and clear representation of the results. The methods are freely available on github

Minor comments

Line 6 – not sure reference 2 is the most appropriate. Reviews by Suttle et al (or many other reviews) better explain the importance of viruses in biogeochemical cycling

P21- what was used to extract the kmers?

P21 L 406 – table number missing

L410 – where any cutoffs used for domain identification ? Please state them if so

Minor issues in the references, I suspect a reference manager issue

Bacterial names to be italicised eg Synechococcus [5]

Inconsistent capitalisation of titles [7]

Provision of DOIs on some references [10, 13 ]

Incorrect citation of HMMER [40] cite the paper not the user guide

**Have all data underlying the figures and results presented in the manuscript been provided?**

Reviewer #1: Yes

Reviewer #2: Yes

Reviewer #3: Yes

PLOS authors have the option to publish the peer review history of their article (what does this mean?). If published, this will include your full peer review and any attached files.

Reviewer #1: No

Reviewer #2: Yes: Simon Roux

Reviewer #3: No

---

## [Decision Letter · Decision Letter 1]

10 Mar 2020

Dear Prof. Robertson,

Thank you very much for submitting your manuscript "Predicting host taxonomic information from viral genomes: a comparison of feature representations" for consideration at PLOS Computational Biology. As with all papers reviewed by the journal, your manuscript was reviewed by members of the editorial board and by several independent reviewers. The reviewers appreciated the attention to an important topic. Based on the reviews, we are likely to accept this manuscript for publication, providing that you modify the manuscript according to the review recommendations.

Sincerely,

Morgan Langille, PhD

Associate Editor

PLOS Computational Biology

Virginia Pitzer

Deputy Editor

PLOS Computational Biology

[LINK]

Reviewer's Responses to Questions

**Comments to the Authors:**

Reviewer #2: The authors provide a revised manuscript with a number of modification and additional work, that I found overall interesting and well done. I am still not entirely in agreement with the author’s interpretation, e.g. I do not believe that any AUC>0.5 reflects a meaningful and useful host adaptation signal, however the presentation by the authors of their analysis and results remains fair and now includes the proper caveat, so a reader (even non-specialists) can form their own opinion.

The manuscript however, especially the new texts and figures, must be carefully reviewed for grammar and typographic errors, and to avoid internal contradictions. I have noted a few instances below, but these are likely not exhaustive, and the manuscript really needs to be carefully checked. The most important aspects for me are some parts that seem to contradict each other, e.g. the authors state p. 29: “We have limited this study to using k-mer composition of the sequences.”, and then p. 30: “In conclusion, our results show that features derived from all four representations of viral genomes [i.e. including domains] are predictive across the host tree”. This is the kind of issue that makes the current version of the manuscript sometimes hard to follow, and requires a thorough and rigorous proofread. The authors must also be more rigorous in their presentation of the figures and tables: each display item must be accompanied by an extended legend briefly summarizing the work presented and explaining all the different acronyms used. Said simply: a reader should be able to understand each figure and table without having to read the main text, which is not the case at the moment.

p. 2: “or machine learning approaches as described below.”: I disagree: machine learning are one approach used to interpret the different features mentioned by the authors, but not an alternative to e.g. “k-mer composition comparison” or “co-abundance between virus and host”

p. 2: “Machine learning approaches offers reference- and alignment-free alternative”: I also disagree: machine learning approaches rely 100% on a training set, which is the equivalent of “references” in other approaches such as prophage detection

Table 1: Please specify the meaning of “PC” in the table legend.

p. 10: “PC_5 is from the physio-chemical sequences with k-mers of length 5”. Why do the authors only mention “PC_5” instead of explaning the different acronyms (AA / PC) and then mention that the number corresponds to the k-mer size ?

same p. 16: “AA_2, is from the amino acid sequences with k-mers of length 2” ?, and for Fig. 8 legend and Fig. 9 legend

p. 23: “The resulting signal will be more relevant to diverging sequences and thus this signal is more likely to have been removed in the holdout datasets”

XX → I am confused by this statement: wasn’t this the exact opposite of the holdout idea, i.e. holding out the dataset mean that most of the signal should come from diverging sequences ?

Said otherwise, a drop of AUC in holdout datasets would suggest that the signal was mostly originating from phylogenetic relationships between viruses (i.e. similar viruses), rather than host-specific features conserved across unrelated viruses infecting the same host ? I think this should be clarified in the text as this is an important point in this type of analysis.

l. 24: “Siphoviradea” should be “Siphoviridae”

Fig. 10 needs a larger legend than “Combined kernel classifiers.”

p. 27: “Reassortment on co-infection not only means that these viruses are highly diverse but gives them a mechanism to share genome segments from multiple hosts .” I am not following the logic here: co-infection means that viruses infect the same host, so how would this lead to higher rate of exchange between viruses infecting multiple hosts ? Please clarify (or remove)

p. 28: “oligio-nucleotides” should be “oligo-nucleotides”

p. 29: “they are wrongly labeled false” is unclear, please rephrase

p. 29: “may be more to do” should be “may have more to do”

p. 29: “diverses” should be “diverse”

p. 29: “all available host labelled data available” should be rephrased to avoid the repetition

p. 29: “We have limited this study to using k-mer composition of the sequences” I am confused by this statement, since the authors also use PFAM domains, which seems to provide a similar (if not better) signal than k-mer in some situations ? (e.g. Fig. 2 panel A - “All”) ?

**Have all data underlying the figures and results presented in the manuscript been provided?**

Reviewer #2: Yes

PLOS authors have the option to publish the peer review history of their article (what does this mean?). If published, this will include your full peer review and any attached files.

Reviewer #2: Yes: Simon Roux
---

## [Editor Report · Decision Letter 2]

21 Apr 2020

Dear Prof. Robertson,

We are pleased to inform you that your manuscript 'Predicting host taxonomic information from viral genomes: a comparison of feature representations' has been provisionally accepted for publication in PLOS Computational Biology.

Best regards,

Morgan Langille, PhD

Associate Editor

PLOS Computational Biology

Virginia Pitzer

Deputy Editor

PLOS Computational Biology

---

## [Editor Report · Acceptance letter]

13 May 2020

PCOMPBIOL-D-19-01489R2 

Predicting host taxonomic information from viral genomes: a comparison of feature representations

Dear Dr Robertson,

I am pleased to inform you that your manuscript has been formally accepted for publication in PLOS Computational Biology. Your manuscript is now with our production department and you will be notified of the publication date in due course.

With kind regards,

Laura Mallard
